# Consensus on the Best Practice Guidelines for Psychomotor Intervention in Preschool Children with Autism Spectrum Disorder

**DOI:** 10.3390/children9111778

**Published:** 2022-11-19

**Authors:** Adriana Frazão, Sofia Santos, Ana Rodrigues, Teresa Brandão, Celeste Simões, Paula Lebre

**Affiliations:** 1INET-md, Instituto de Etnomusicologia, Centro de Estudos em Música e Dança, Faculdade de Motricidade Humana, Universidade de Lisboa, 1499-002 Lisboa, Portugal; 2UIDEF, Instituto da Educação, Faculdade de Motricidade Humana, Universidade de Lisboa, 1499-002 Lisboa, Portugal; 3Centro de Estudos em Educação, Faculdade de Motricidade Humana, Universidade de Lisboa, 1499-002 Lisboa, Portugal; 4ISAMB, Instituto de Saúde Ambiental, Universidade de Lisboa, 1499-002 Lisboa, Portugal

**Keywords:** consensus guidelines, nominal group technique, Delphi survey technique, children, health promotion, psychomotor intervention, autism spectrum disorder

## Abstract

Psychomotor intervention has been used to promote development by the enhancement of psychomotor and socio-emotional competence. However, studies with high-quality evidence, describing psychomotor-intervention processes and outcomes are scarce. Therefore, we aimed to generate expert consensus regarding psychomotor-intervention guidelines to support psychomotor therapists through the design and implementation of interventions for preschool (3–6 years old) children with autism spectrum disorder (ASD). A formal consensus process was carried out, using modified nominal group (phase I) and Delphi survey (phase II) techniques. We recruited 39 Portuguese experts in psychomotor intervention with preschool children with ASD in phase I. Experts participated in at least one of the five online meetings, discussing themes (e.g., objectives, methods, strategies) concerning psychomotor intervention with preschool children with ASD. A deductive thematic analysis from phase I resulted in 111 statements composing round 1 of the Delphi survey. Thirty-five experts completed round 1, and 23 round 2. The experts reached a consensus (agreement > 75%) on 88 statements, grouped under 16 sections, (e.g., intervention source, general setting, intended facilitation-style), reflecting generic psychomotor-intervention guidelines. Consensus guidelines may be used to support transparent and standard psychomotor interventions, although further studies should be undertaken to determine their efficacy.

## 1. Introduction

Psychomotor intervention, also known as psychomotor therapy or psychomotricity, is a specific intervention modality focused on body movement-centered approaches for mental health, facilitated by a psychomotor therapist [1,2]. Psychomotor intervention is focused on an integrated perspective of the human being and the importance of body and movement on the development of an individual’s competency within a psychosocial context, where body and mind dimensions interact with each other [2]. Psychomotor intervention provided across healthcare, social, and educational settings has been recognized in several European, Central and South American countries. Depending on the purpose of the intervention, preventive or therapeutic approaches are used with children with typical development and with specific health needs. Psychomotor interventions focused on preventive approaches have shown positive results in the development of preschool children [3,4,5,6,7] as well as therapeutic approaches which have demonstrated benefits for children aged 3–6 years with special healthcare needs, namely, with autism spectrum disorder (ASD) [8,9,10].

Considering that motor delays are present among the earliest markers of ASD, with a prevalence reported in more than 80% of children diagnosed [11,12], and that motor skills during play with peers and caregivers are critical to developing social interactions and engagement [13,14], psychomotor interventions can be regarded as beneficial for children with ASD. However, with a broader scope, psychomotor as well as other movement-based practices, i.e., *interventions that that use physical exertion, specific motor skills/ techniques, or mindful movement to target a variety of skills and behaviors* [15] (p.4024) have insufficient evidence, although positive effects were identified [15,16,17]. Review studies on the effects of psychomotor intervention with children reinforce the scarcity of high-quality evidence [10,18] specifically with children with ASD, since most studies are case-control studies or case reports, with insufficient (or even contradictory) results and with no clear or detailed description of the implementation of practices provided [10,18]. As with other health professions [19,20,21], decisions about psychomotor intervention practices should be evidence-based, so that the best practices are implemented. Since there is limited evidence guiding professionals on how to provide such high-quality interventions, it is necessary to identify these specific features.

So far, only general guidelines on non-pharmacologic evidence-based interventions have been published to support professionals working with children with autism and their families [22,23,24,25]. International recommendations include individualized programming for each child; ongoing program evaluation; interventions emphasizing attention, imitation, communication, play, social interaction, regulation, and self-advocacy; a highly supportive environment; predictability and structure in the sessions; functional analysis of possible behavior problems; and family involvement. In Portugal, individualized intervention according to the child profile and contexts with both therapeutic and educational components are recommended [26], although no specific guidelines for psychomotor intervention has been provided.

When scientific evidence is either insufficient or conflicting, consensus protocols, where experts come to an agreement on recommendations, provide an alternative approach to develop guidance, in order to enhance implementation and fidelity among professionals. The Consolidated Standards of Reporting Trials for social and psychological interventions (CONSORT-SPI 2018) is an example of an instrument that guides behavioral and social researchers to report studies transparently [27].

The World Health Organization (2014) has recognized the need to use formal consensus methods in the development of clinical practice guidelines, and often uses the NGT and Delphi technique to identify stakeholders’ priorities for integration into healthcare recommendations [28,29,30,31]. The nominal group technique (NGT) and the Delphi technique are used as a consensus approach for integrating knowledge [32]. In this study, we aim to develop consensus guidelines regarding specific characteristics of psychomotor interventions for preschool children with ASD.

## 2. Methods

### 2.1. Selection and Recruitment of Participants

In the present study, academic and professional experts in psychomotor intervention with preschool children with ASD. were recruited. The ‘snowball’ technique was used to select the group of experts. In total, 70 Portuguese experts were invited by email, and provided them with information about the aim, conditions, and procedures of the study. First, the invited experts filled out an online form to gather their details regarding academic/professional experience, personal data and availability to meet and to complete online surveys.

The inclusion criteria were:
(a)Have at least a bachelor’s degree in Psychomotor Therapy.


AND
(b)Be a psychomotor therapist with more than three years of experience in psychomotor intervention in different contexts in Portugal, working either individually or in groups (using therapeutic or educational approaches).

OR
(c)Be a researcher or university professor with scientific/professional expertise in preschool children with ASD, in psychomotor intervention.

The experts’ demographic characterization is presented in the following Table 1.

### 2.2. Procedures

The study was approved by the Faculty of Human Kinetics Ethics Council (reference number 48/2021 on 29 November 2021). Experts did not receive any payment for participation. Written informed consent to take part in this study was obtained from all participants.

The most common approaches to consensus development in healthcare and clinical fields are: the nominal group technique; the Delphi technique; and the consensus-development conference [33]. The nominal and Delphi technique chosen involve a group of ‘experts’ generating ideas and determining priorities [32]. Both are equally effective, and more effective than conventional interacting groups [30]. This study design comprised a combination of modified nominal group and Delphi techniques (see Figure 1) [32].

#### 2.2.1. Phase I: Nominal Group Technique (NGT)

The NGT, a formal structured consensus-development method to integrate experts’ points of view in order to gain consensus, followed recommendations proposed by McMillan and colleagues [32,34]. First, researchers informed the participants of the aim and importance of group-expert meetings, and then online meetings on the Zoom platform took place in groups (*n* = 5) of 5 to 7 participants, over a 3-month period from April to June 2021. Two facilitators were responsible for the experts’ meetings: a primary facilitator leading the meetings, and an independent researcher responsible for writing down the experts’ ideas. Upon consent, meetings were recorded, transcribed and coded.

Regarding the meetings, initially a brief study-overview was presented. In each meeting, a research question was presented and discussed for 60–90 min. The questions were selected upon an initial scoping review of psychomotor intervention practices with preschool children with ASD [10], which included: (1) objectives and primary results, (2) theories and models, (3) methods, (4) environmental arrangements, and (5) strategies (see all questions in Appendix B, Table A1). Each meeting followed the four stages of NGT [32,34]: silent generation, round robin, clarification, and ranking.

In that last stage, instead of allowing participants to vote on their preferred ideas as in the traditional NGT technique, participants were invited to propose categories and then rate them, from the generated list. The rating on a scale from 1 (not at all) to 5 (extremely essential or important) concerned how essential or important the ideas were in each category. This rating process was completed individually on an online form, without any group discussion. The ratings results were given to the experts by email after each meeting.

#### 2.2.2. Delphi Technique

After the content-analysis of transcriptions and rated categories in the NGT, three researchers (A.F., S.S. and P.L.), drafted consensus statements for the Delphi survey. The guidelines’ survey statements followed the NICE writing principles [35], and were organized into sections and themes (see Table 2), adapting a checklist designed to improve the reporting of group-based interventions [36]. This checklist included the main elements of an intervention program, according to the CONSORT-SPI 2018 [27].

The Delphi survey, with two rounds, was conducted over a 4-month period from February to May 2022. The anonymization of responses was guaranteed. Before each round, experts received an email with a link to access the survey. Each statement was accompanied by a 9-point hedonic scale asking for the indication of the agreement level (with 1 indicating ‘completely disagree’ and 9 ‘completely agree’). In addition, respondents were encouraged to give open-ended comments for each statement. Since the NGT was used in the first step, the Delphi technique skipped the first traditional rounds. Instead, each round was analyzed on completion, with the results contributing to the content of the subsequent round. Only experts who completed the first survey round were invited to participate in the next round [31]. Only statements with a median score greater than 7 were accepted. A second round of the survey was provided, since less than 75% of given statements scored below 7 [31,37]. The Delphi survey process terminated when consistency responses between rounds achieved 85%, i.e., more than 85% of the same statements showed consensus in two rounds, reflecting a substantial majority on topics about psychomotor intervention practices with preschool children with ASD.

### 2.3. Data Management and Analysis

#### 2.3.1. Phase I

In the NGT, both qualitative and quantitative (numerical rankings) data was obtained.

Excel spread sheets from *Google forms* were used to gather experts rankings. Descriptive data was used for the results. Each NGT focus group was recorded via the *Zoom* platform, and written field notes were taken. The traditional approach of reading the transcribed audio recordings and listening to the recordings for verification of the text, was used.

#### 2.3.2. Phase II

The statements of the Delphi survey emerged from the content analysis of phase I. Based on the output of the NGT meetings, a thematic text analysis for identifying, analyzing and reporting patterns (themes) across transcriptions of the meetings was conducted, using MAXQDA software. The thematic analysis was chosen since it follows a theoretical (deductive) approach based on the literature review [38]. This method is not associated with a pre-existing theoretical framework [38]. Data familiarization, generation of initial codes, searching the data for themes, reviewing themes, and reaching agreement for defining themes, was conducted. The initial themes (codes) were generated by one researcher (A.F.). Two researchers (S.S. and P.L.) independently reviewed the themes where an agreement was not reached. Concerning the theme “change techniques”, the play types followed a typology defined in *Tool for Observing Play Outdoors* [39]. The themes in the transcriptions that were rated higher and emerged more frequently across meetings were the basis used to generate the statements. In the Delphi technique, the frequency of responses for each statement (item) was calculated. If the statement had a rating of 7–9 among at least 75% of experts, this meant that consensus had been reached. The statements that did not complete this criterion were excluded. The included statements were forwarded to the subsequent round for re-rating. From round one to round two, four researchers (A.F., C.S., S.S. and P.L.) analyzed the data and reformulated or included statements, considering the experts’ comments.

## 3. Results

The experts’ mean age (M = 37) ranged from 26 to 68, most being female (98%). The majority of respondents were practitioners working in the field for at least five years, predominantly in a clinic and/or home context.

### 3.1. Phase I

The first NGT meeting involved 39 Portuguese experts, where 35 were psychomotor therapists and 4 researchers in psychomotor intervention (Table 1). Approximately 80% of them participated in all meetings.

The NGT was used to inform guideline development through the generation and prioritization of topics, with experts being asked questions such as, “What theories or models support psychomotor intervention with children with ASD (3–6 years)?”; “What are the strategies you use most (in individual and/or group intervention)?” (see Appendix B). Following the rating of each item, experts were asked to select their “top five”, in terms of relevance. From the first meeting, and after the independent analysis of the responses concerning the objectives and primary results of the psychomotor intervention, a list of 19 categories were identified. The most relevant/priority categories proposed were: social interaction, (non-verbal) communication, playfulness, self-regulation, and psychomotor skills. From the second meeting, a list of 29 categories concerning theories and models that support psychomotor intervention were obtained. The conceptualizations proposed which were considered as the most relevant/priority were: DIR Floortime model, Applied Behaviour Analysis approach, Early Start Denver Model and TEACCH method. From the third meeting, 19 categories concerning the type of activities in psychomotor intervention sessions were considered as the most relevant/priority: free play, perceptive/sensory activities, and bodily-movement activities. From the fourth meeting, 26 categories were related to the session structural aspects, where the most relevant/priority were: materials and space organization, parents’ participation; stakeholders’ communication; session length and session moments. In the final meeting, 31 categories of strategies used in psychomotor intervention were: attention to the interests of the child, child–therapist relationship, complexity of activities proposed, and feedback and reinforcement.

After deductive thematic analysis of the NGT meetings (i.e., notes, categories of ideas rated, and audio-recordings transcriptions), 74 subthemes were obtained. These subthemes were classified under 19 themes (Table 2), in order to organize the statements by sections for phase II. Each statement intended to summarize the most repeated ideas in the coded transcriptions, which were then considered as the most relevant for each NGT meeting. These statements were the basis for writing the statements for the Delphi survey.

### 3.2. Phase II

Figure 2 presents the flow of experts and statements through the two rounds of the Delphi survey. 35 Portuguese experts (34 psychomotor therapists and one researcher) completed round 1, and 23 completed round 2 (65%). All experts had a minimum of 5 years’ experience in psychomotor intervention or research. Descriptive statistics of the 111 statements is presented by percentage of agreement obtained (percentage of experts’ rating between 7 to 9; see Appendix A round 1). The percentage of agreement ranged from 31.4% (statement 56) to 100% (statements 7, 17, 20 21, 26, 66, 84, 100, 107). For 26 statements, consensus was not reached. Regarding the sequencing of sessions and activities during the sessions, the statements showed that no consensus was reached, and thus these sections were excluded from round 2. Experts provided 79 free-text commentaries or suggestions regarding mainly the clarity and relevance of the statements. Based on a thematic analysis of this qualitative data, three new statements were generated, twelve were reformulated and five (the text of the statement was maintained) were moved to another section (see Appendix A).

In round 2, the 94-item anonymized survey was completed by 23 experts. The percentage of agreement ranged from 65.2% (statement 10) to 100% (statements 4, 5, 11, 14, 15, 16, 18, 21, 50, 51, 52, 53, 66, 67, 86, 87). A tendency of increase in agreement across experts was observed. Six statements were excluded from section E, “Frequency of sessions”. Experts provided 53 commentaries related to the need to consider each child’s individuality. Thematic analysis generated no new items, and termination criteria for the Delphi survey were fulfilled. Experts reached consensus for the inclusion of 88 statements, grouped under 16 sections, reflecting guidelines for developing psychomotor intervention with preschool children with ASD.

Table 3 presents the 42 consensus guidelines that reached 95% or more of agreement among experts in the last round. The complete list of 88 guidelines is available as Appendix A. The sections with guidelines that met a stable consensus in both rounds with 95% or more of agreement were: intervention source, venue characteristics, change mechanisms, change techniques, strategies during the sessions, therapists delivering the sessions, and intended facilitation style (Table 3).

## 4. Discussion

This study addressed the need of consensual guidelines within psychomotor intervention with preschool children with ASD. As such, this study used aa NGT and a Delphi survey to achieve expert consensus, aiming to better describe psychomotor intervention practices and generate more clear and effective interventions among psychomotor professionals in the future.

In phase I, 39 Portuguese experts, including psychomotor therapists working predominantly in clinical contexts of intervention (46% response rate) and researchers in psychomotor intervention (10% response rate), participated in a modified NGT process. In phase II, 23 of those experts completed two rounds of a Delphi survey (65% response rate), quantitively rating and qualitatively refining survey statements. No major changes were made between rounds, and consensus on the most important/highest priority items was reached.

With a border scope, the consensus guidelines obtained in this study are in line with the generic international guidelines for intervention with children with ASD [22,23], such as guideline 4. (individualized intervention), 26. (focus on individual strengths), 46. (objectives of intervention), 51. (supportive environment) 74. (stakeholders involvement). Most guidelines identified in this study are specific and detailed regarding the characteristics of psychomotor intervention, namely the guidelines describing change techniques and how they should be facilitated, and materials and intended facilitation, and therefore these are not mentioned in the international documents. Regarding the change techniques, the type of play used in psychomotor intervention is the major focus of movement play, structured opportunities, fine-manipulation play, expressive play, symbolic play, conventional games, constructive exploration, and sensory-exploration play. The 88 consensus guidelines were grouped under 16 sections. The most highly rated items in terms of relevance refer to the fact that psychomotor interventions can occur individually or in groups, should guarantee the child’s safety and autonomy, follow the child’s motivations, and create opportunities to develop competencies, so that sessions create a playful climate. Regarding psychomotor therapists’ competencies, several consensus statements focused on the quality of relationship as well routines and instructions during the sessions, under ongoing supervision and continuous training. This allowed us to consider that these are priorities for a clear definition of psychomotor intervention. All respondents agreed that these guidelines should have sufficient flexibility to be adapted according to the child and group characteristics. Furthermore, these guidelines seem to reinforce the multidimensional and interdisciplinary feature of psychomotor interventions, which should consider the components related to intervention design, intervention content (including theories that support intervention), participants, and facilitators, which interact with each other and impact final outcomes. As reflected in the consensus guidelines, psychomotor therapists use behavioral and educational approaches which are recommended for ASD [26]. This study adds to the scientific literature a tool to guide and support the psychomotor therapists’ practices, planning interventions according to the features recommended by adopting the methods, techniques, and strategies necessary to take into account in interventions with preschool children with ASD.

A comparison with other studies that used a similar methodology with the aim of developing intervention guidelines, shows that the combination of NGT and the Delphi technique are found in medical research [40,41,42]. Nevertheless, this combination was not found in therapeutic studies. Development of healthcare practice recommendations usually used methods based on systematic reviews and meta-analyses [43]. The specific research question and the methods are novel in psychomotor-intervention research. The combination of the two techniques was used to answer our research question, following a prescribed set of procedures [32,41]. A Delphi survey preceded by an NGT process made for a complex data analysis that was necessary to guarantee that guidelines were based on the heterogeneity of experts’ opinions. However, the use of the RAND/UCLA method (a hybrid of the Delphi and NGT) is suggested, to simplify the consensus process [41] in a future replication of this study.

Regarding the limitations of the study, the size and composition of the panel of experts may influence the quality of the data, and whether the judgments are accepted and considered feasible, especially if the number of experts is rather low [31]. Since this study included only experts from Portugal, it may be considered that the consensus guidelines identified will require further exploration, by consulting other internationally recognized experts in the psychomotor-intervention field [31,44]. The number and diversity of experts, mainly in the second phase, was lower than originally anticipated; however, the groups of experts were heterogeneous, with a wide range of years of experience and intervention contexts. The composition of the expert panel would benefit if the number of years of experience in psychomotor intervention and the number of ASD children accompanied in the last two years were higher. Thus, it is our opinion that a balanced number of therapists and researchers, between therapists working in therapeutic and educational contexts and the inclusion of experts from different countries, will improve the contextual and geographic representation in future studies. Nonetheless, the expert sample-size of this study may also be regarded as a strength, since more than double the proposed size limit of 12 was achieved.

Another limitation of this study was the fact that categories of ideas rated in the NGT process were generated at the end of each meeting while the participants were providing their suggestions. This process could be improved with a classification of categories created with more time, namely after the ideas gathered from experts allowing a deeper reflexion of the summarized and described ideas. Moreover, since the statements during the Delphi survey process were drawn from a content-analysis procedure, the subjectivity of the authors’ experience and knowledge (A.F., S.S. and P.L.) could have influenced the final consensus guidelines.

In phase I, group feedback was provided after each meeting. In phase II no feedback (individual or group) was provided to participants between rounds, contrary to what is considered to be a feature which is regarded as a strength of consensus methods [43]. In both phases, experts commented on key-terms definitions (e.g., non-directive vs. directive approach, structured vs. unstructured activities, play vs. game, guidelines…). A clear a priori statement of the definition of concepts ensured that assessment statements by experts started from a common understanding of key concepts and definitions, to avoid subjective perspectives and biases [44]. However, this is the first study, as far as we know, that attempted to systematically achieve consensus within the psychomotor-intervention field, where a transparency of processes was guaranteed, and all the study was done in accordance with internationally recognized guidelines (CONSORT-PSI 2018). These guidelines fill the gap in evidence based on a consensus of the experience of experts in this area, supporting the future development of psychomotor-intervention efficacy studies. It is recommended that a consensus-based approach drawn from systematic reviews on the topic under study, is conducted.

The authors understand first that further work will be required to test the usefulness and applicability of the proposed guidelines by psychomotor therapists currently undertaking intervention with children with ASD in Portugal and in other countries. Secondly, a replication of this study, including psychomotor-intervention experts working in educational contexts from different countries, a multidisciplinary clinician group, and service-user representatives, is recommended. Thirdly, further developments of this research may include the study of these guidelines for children with typical development, considering the role of peers that support inclusion within natural contexts. Finally, more research is recommended regarding the impact of psychomotor interventions for preschool children with ASD. Because scientific evidence regarding psychomotor intervention for children with ASD is scarce [7,10], this study will contribute to guiding the planning and reporting of psychomotor interventions that is necessary to guarantee the quality of intervention.

## 5. Conclusions

The current study engaged experts to develop and reach consensus guidelines addressing the transparency in psychomotor intervention and related research. This study is an important first step within the psychomotor-intervention field since the findings provide a summary of specific features of psychomotor interventions with preschool children with ASD. Results gave insights into how practices are, and should be, conducted by psychomotor therapists. The consensus-based approach resulted in 88 guidelines, where 42 reached more than 95% of agreement. The aim of these guidelines is to contribute to evidence studies by tailoring the coverage and challenges to define psychomotor intervention as a comprehensive intervention. We recommend the use of these guidelines (and their sections) to standardize the planning and reporting of psychomotor-intervention programs that can be more easily replicated, disseminated, and researched. Given the potential impact of psychomotor interventions in the development of preschool children with ASD, it is imperative to conduct high-quality research in the future. The issues identified in the present study can inform the research and practical agenda in the psychomotor therapy field. We suggest a broader representation of experts, to ensure these guidelines retain relevance to all stakeholders involved in the growing national and international field of psychomotor intervention. Furthermore, this study could serve as a model for generating consensual guidelines within psychomotor therapy in other target groups.

## Figures and Tables

**Figure 1 children-09-01778-f001:**
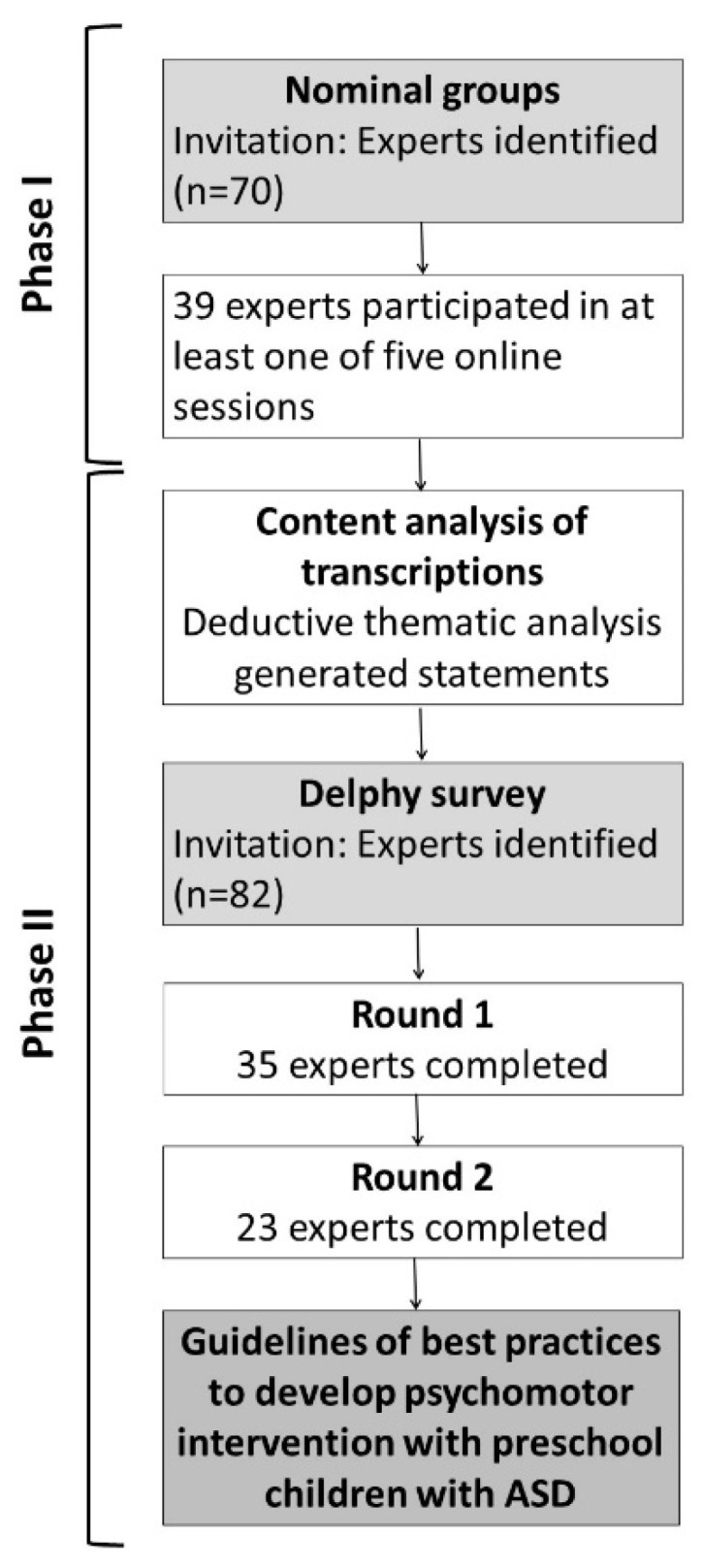
Phases of the consensus approach and experts involved.

**Figure 2 children-09-01778-f002:**
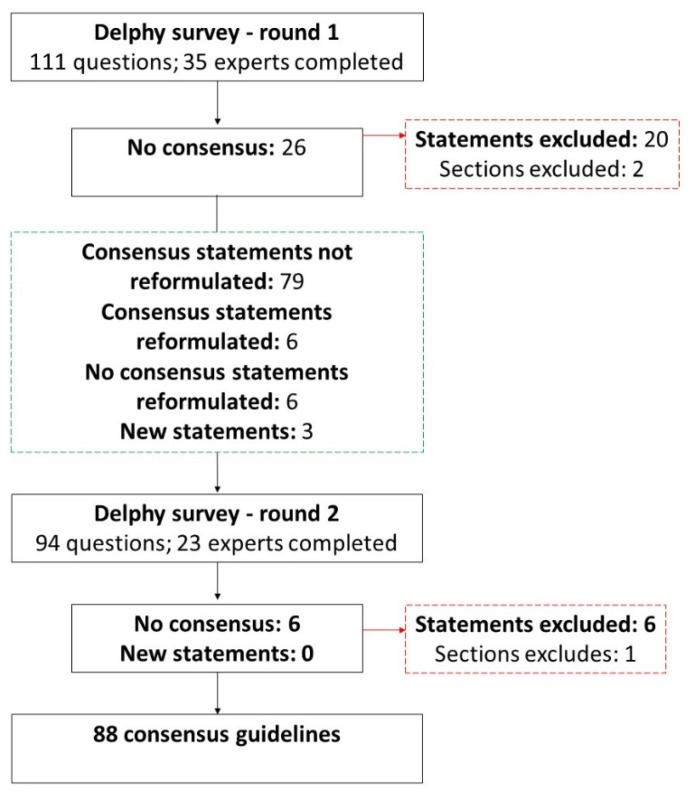
Delphi survey process.

**Table 1 children-09-01778-t001:** Experts demographics.

	NGT Process	Delphi Survey–Round One
Age	M	M [min, max]
Years	37 [26, 67]	37 [26, 68]
Gender	*n*	*n*
Female	38	5
Male	1	0
Current profession	*n*	*n*
Psychomotor therapist	35	34
Professor/researcher	4	1
Working experience	M [min, max]	M [min, max]
Years of experience	13 [5, 45]	15 [5, 45]
Number of children with ASD accompanied (last 2 years)	8 [0, 30]	9 [0, 30]
Number of children accompanied in group settings with or without ASD (last 2 years)	7 [0, 25]	3 [0, 25]
Context of intervention (predominantly)		
Clinic	18	18
Home	8	8
Community teams/association	3	2
School	3	2
Hospital	3	2
Others	4	2

**Table 2 children-09-01778-t002:** Themes (sections) obtained in the NGT.

Intervention Design	Intervention Content
Intervention source General setting Venue characteristics Length of sessions Frequency of sessions Duration of the intervention	G. Change mechanisms or theories of change H. Change techniques I. Session content J. Sequencing of sessions K. Materials L. Activities during the sessions M. Strategies during the sessions N. Methods for generalization O. Methods for checking fidelity of delivery P. Methods for checking social validity
**Participants**	**Facilitators**
Q. Participants during the sessions	R. Professionals delivering the sessions S. Intended facilitation style

**Table 3 children-09-01778-t003:** Key consensus recommendations for psychomotor intervention with preschool children with ASD.

**Sections**
**A.** **Intervention source** 1. Psychomotor intervention facilitates play opportunities and movement-based experiences for the child to explore, experience and feel competent in the bodily interaction with the context, in educational and therapeutic approaches.
**B.** **General setting** 3. Psychomotor intervention can take place in different spaces (home, swimming pool, equestrian center, outdoor spaces) depending on the child’s needs and objectives. 4. Psychomotor intervention can occur individually or in groups, where individual interventions allow the strengthening of the therapeutic relationship and the reinforcement of skills with individualized strategies.
**C.** **Venue characteristics** 5. The space should guarantee the child’s safety and autonomy.
**F.** **Duration of the intervention** 11. The duration of the intervention depends on the child’s profile, the work carried out in the family and school context, and the articulation between stakeholders and resources.
**G.** **Change mechanisms or theories of change** 12. The preschool period is a sensitive period regarding environmental stimuli related to motor and social development. 14. The child’s motivations (interests) are the starting point for developing competencies. 15. Playfulness promotes active, interactive, and pleasurable learning. 16. The therapeutic relationship should be guided by active listening, empathy, authenticity, and trust. 17. Safety is provided by a structured space and time, and by a welcoming and affectionate climate. 18. Sensorimotor experiences promote sensory information processing and processing related to planning, learning and control. 20. The movement of the body produces physiological and emotional responses that are regulated according to the interaction with the social and physical environment, and is systematized in the implicit memory responsible for the pleasure in action and the relationship with the other. 21. Body (and self) awareness influences the ability to feel and understand the emotional state of the other and, therefore, influences interpersonal relationships. 23. The use of expression-oriented activities/strategies and action-oriented activities/strategies provides a comprehensive and integrated response to a child’s development.
**H.** **Change techniques** 26. The techniques applied in the context of a gymnasium or therapy room should be promoted in the form of different types of play, selected according to the characteristics and individual interests of the child and the specific objectives of the intervention. 27. The type of games played in the first intervention sessions should be based on the child’s spontaneity and/or interests. 31. Fine-movement play: drawing, painting, modeling, fine manipulation and writing materials in different planes and spatial dimensions should be facilitated. 32. Movement games, massage and imitation of simple movements involved in a story or song can be used, to provide the perception of the contrast between contraction and relaxation states. 33. Expressive play: different music and rhythms creating sequences of movements, can be used. 34. In expressive play, different types of movements, positions and body postures that vary between states of movement and immobility should be facilitated. 38. Symbolic imaginative play: objects, actions, or ideas that symbolize something should be facilitated. 40. Conventional-rules games, (i.e., playing football): knowledge of rules for “know-how” enables the mobilization of skills of perception, memorization, planning, organization, sequencing and association. 42. Constructive exploration play (i.e., building a tower): construction materials should be available. 43. Sensory exploration play: materials with different sensory properties (visual, auditory, tactile, proprioceptive, vestibular) should be facilitated. 44. Sensory exploration play: experiences of therapeutic/affective touch, breathing, body restraint and passive mobilization using mediation materials, should be facilitated.
**M.** **Strategies during the sessions** 50. The child’s interests, curiosity, desire for exploration and initiative, should be followed to promote active engagement and learning. 51. The manipulation of involvement through the organization of space, the selection of materials and the positioning of the psychomotor therapist’s body should guide the exploration of the child’s possibilities of action, introducing or removing elements. 52. Verbal instruction should be clear, simple/short and directive, of the type associated with gestures or demonstration. 53. Visual or gestural prompts should be used when there is a need to guide the child’s response to act in accordance with the objective, during instruction or execution of the activity. 60. The total or partial physical and verbal assistance that leads to the achievement of the objective in the activity should evolve from a greater to a lesser intensity. 64. The psychomotor therapist should create challenges between the child and what he wants to do, to promote the child’s interaction/communication when there are situational conditions and a trusting, therapeutic relationship. 65. Anticipating what will happen through planning the session structure with the child and establishing routine moments should happen to help the child manage their emotions and behaviors, through drawing, images, verbalization, repeated actions and definitions of spaces and materials. 66. The description of observable emotions and behaviors of the child should be used whenever there is a need to help the child to self-regulate. 67. The therapist’s tonic receptiveness should induce the child’s emotional safety, interaction, and communication, through the psychomotor therapist’s posture, looking from the child’s eye-level, and being in synchrony with the verbal and non-verbal communication and tone of voice.
**R.** **Therapists delivering the sessions** 86. The psychomotor therapist should guarantee commitment and professional competence in monitoring the intervention centered on the child and the family. 87. The psychomotor therapist should resort to supervision/tutoring and continuous training based on scientific evidence.
**S.** **Intended facilitation style** 88. The psychomotor therapist should provide conditions for the child to express and practice their abilities (strengths) to reduce their difficulties (weaknesses). 90. The psychomotor therapist should actively observe or participate with the child. 91. The psychomotor therapist’s approach should be directive or non-directive, according to the characteristics of the children and the therapeutic goals. 92. The psychomotor therapist should establish an authentic and safe therapeutic relationship. 93. The limits/rules actively defined by the group should be known to all group elements, whenever necessary. 94. The facilitation of group activities should provide for the participation of all group elements, making the necessary adaptations so that all children can demonstrate their abilities, and welcoming different forms of performance.

## Data Availability

Not applicable.

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
