# Peer review of "Consensus on the Best Practice Guidelines for Psychomotor Intervention in Preschool Children with Autism Spectrum Disorder"

_children, 2022, doi:10.3390/children9111778_

Round 1

Reviewer 1 Report

Very interesting work both in terms of methodology and results achieved.

1. The Authors rightly point out that it is necessary to continue this research treating the current one as preliminary research. It is certainly a step towards recommending proven therapeutic interventions. This is especially important for practitioners in clinical work but also for parents who are looking for effective activities for their children with autism spectrum disorders.

2. It would be interesting from a practical as well as scientific point of view to test the effectiveness of the prepared criteria in longitudinal studies - how children function in subsequent stages of their lives. Going further - how adults on the autism spectrum see their therapy, which of the selected criteria were relevant to them. But these are ideas for further research.

3. The article was written with a lot of details, detailed descriptions of the whole procedure. In my opinion, the descriptions could be shortened somewhat.

4. A critical analysis of the criteria selected during the study was missing. Perhaps the procedure itself should be changed, simplified, enriched? If the work was supplemented with such a passage, there is value would be greater.

Author Response

Dear Reviewer, 

We would like to thanks the reviewers for all comments and recommendations, which were very important and have substantially strengthened the article. After reading very carefully all comments, critiques and recommendations we are able to say that the changes suggested by the reviewers were made.

Answer to comments:

1. and 2.: Thank you for your positive remarks; as well as for the recommendations for research.

3.: Noted and corrected; according to reviewer comments some passages in the text were deleted and other shortened.

e.g. (line 139):

Each meeting followed the four stages of NGT [32], [34]: silent generation, round robin, clarification and, ranking.

A critical analysis of experts’ inclusion criteria and criteria for selected procedures was reinforced.

e.g. (line 321):

A Delphi survey preceded a NGT process makes a complex data analysis that was necessary to guarantee that guidelines are based in heterogeneity of experts opinions. However, the use of  RAND/UCLA method (a hybrid of the Delphi and NGT) is suggested to simplify the consensus process [41] in a future replication of this study.

The Authors

Reviewer 2 Report

Generally well written. Highly topical topic. Manuscript has an important clinical message. More details about similar studies should be provided.  A several limitation of the study is poor heterogeneity of the panel of experts, only three years of experience in psychomotor intervention as inclusion criteria and the number of ASD children in the last 2 years. The authors should reword these points in the discussion.

Author Response

Dear Reviewer, 

We would like to thanks the reviewers for all comments and recommendations, which were very important and have substantially strengthened the article. After reading very carefully all comments, critiques and recommendations we are able to say that the changes suggested by the reviewers were made.

All comments were noted and corrected; In the text were added more evidences about other studies, but as far as we are concerned this is the first in psychomotor therapy field; The limitation section was reinforced regarding the poor heterogeneity of experts panel; All changes and corrections made are presented in the text with track changes.

e.g. (line 324):

Comparing with other studies that used a similar methodology with the aim to develop intervention guidelines, the combination of NGT and Delphi techniques were found in medical research[40]–[42]. Nevertheless, this combination was not found in the therapeutic studies. Development of healthcare practice recommendations usually used methods based on systematic reviews and meta-analyses [43].

The Authors